# Effect of Different Educational Interventions on Knowledge of HPV Vaccination and Cervical Cancer among Young Women: A Preliminary Report

**DOI:** 10.3390/ijerph19095191

**Published:** 2022-04-25

**Authors:** Yuko Takahashi, Haruka Nishida, Takayuki Ichinose, Yuko Miyagawa, Koichiro Kido, Haruko Hiraike, Hirono Ishikawa, Kazunori Nagasaka

**Affiliations:** 1Department of Obstetrics and Gynecology, Teikyo University School of Medicine, Tokyo 173-8605, Japan; tiamaria198x@gmail.com (Y.T.); nishida.haruka.kg@teikyo-u.ac.jp (H.N.); tichinose@med.teikyo-u.ac.jp (T.I.); m.yuko0201@gmail.com (Y.M.); kidok@med.teikyo-u.ac.jp (K.K.); haruko.hiraike@gmail.com (H.H.); 2Graduate School of Public Health, Teikyo University, Tokyo 173-8605, Japan; hirono@med.teikyo-u.ac.jp

**Keywords:** HPV vaccination, vaccine hesitancy, barriers, health literacy, cervical cancer

## Abstract

The incidence and mortality rates of cervical cancer are rising among young women in Japan. In November 2021, the Japanese Ministry of Health, Labour, and Welfare reinstated the active recommendation for the human papillomavirus (HPV) vaccine, which was discontinued in June 2013 due to reports of adverse reactions, including chronic pain and motor dysfunction, following vaccination. However, vaccine hesitancy among the younger generation remains, and it is essential to identify the barriers in vaccination uptake. Therefore, we aimed to conduct a randomized study using different methods of providing educational contents to improve health literacy regarding cervical cancer and HPV vaccination among female students in Japan. Here, we present the results of our preliminary report and discuss current topics related to HPV vaccination in Japan. Data were collected from 27 female students—divided into three groups: no intervention, print-based intervention, and social networking service-based intervention—using the health literacy scale and communicative and critical health literacy scale. Our primary results indicate that participants’ knowledge and health literacy improved post-intervention. Therefore, medical professionals must provide accurate scientific knowledge regarding routine HPV vaccination and the risk of cervical cancer to young women to improve their health literacy and subsequently increase the HPV vaccination rates.

## 1. Introduction

The incidence and mortality rates of cervical cancer among young women are increasing in Japan [1]. Human papillomavirus (HPV) is a known cause of cervical cancer and is responsible for vaginal, vulvar, and head and neck cancers [2]. The HPV16 and HPV18 subtypes are most commonly associated with cervical cancer. High-risk HPV is detected in almost 100% of cervical cancers, of which approximately 45% and 15% are caused by HPV subtypes 16 and 18, respectively, followed by HPV subtypes 52, 58, 31, and 33. The relative risk of cervical cancer is exceptionally high for women with HPV types 16 and 18, who are 200–400 times more likely to have cervical cancer than HPV-negative women [3]. It is estimated that complete coverage with HPV vaccines in the female population may reduce cervical cancer incidence by up to 90% worldwide [3]. The HPV vaccine was developed to block the route of HPV infection and is an excellent vaccine characterized by the following: (1) the induction of high-titer HPV antibodies in more than 99% of vaccinated persons aged 10–55 years, with titers several to ten times higher than natural antibodies, the absence of non-responders, and the development of HPV antibodies in the absence of non-responders [4,5], and (2) in several countries, large-scale clinical trials in 15–26 year olds have shown that infection with the vaccine subtypes (subtype 16 or 18 or 6, 11, 16, or 18) and the resulting development of moderate cervical dysplasia or other conditions, including cervical cancer, is almost 100% prevented if the patient is uninfected at the time of receiving the vaccine [6]. However, due to reports of adverse reactions such as chronic pain and motor dysfunction after vaccination, the government stopped actively recommending it on 14 June 2013 [7]. Consequently, while the HPV vaccination rate for girls born between 1994 and 1999, who were eligible for vaccination during the period when the public subsidy was introduced, was around 70%, for girls born after 2000, it dramatically declined; the vaccination rate for girls born after 2002 was less than 1% [8].

Although the vaccine coverage rate varies among countries, studies conducted in countries with national HPV vaccination programs have demonstrated clear benefits of mass vaccination in reducing viral prevalence and associated disease burden [6]. In addition, three doses of HPV vaccine are recommended for young women aged 15–26 years [7]. However, in Japan, the three-dose completion rate remains low without government interventions [8], and more importantly, public doubts about the safety of the vaccine remain, impacting vaccination coverage, which cannot improve unless awareness regarding the risk of cervical cancer and its vaccine is raised [9,10]. Despite the merits of implementing HPV vaccination programs for young women, insufficient work has been done to improve this situation worldwide, including in Japan. Although efforts have been made to educate the public about the efficacy and merits of the HPV vaccine, in the United States, the vaccination coverage rate for tetanus, diphtheria, and pertussis vaccines was approximately 80–90% [11], whereas 7.3 million of those eligible for the HPV vaccine were reported to be unvaccinated [5]. Furthermore, there was a decline in HPV vaccine uptake in Ireland owing to increased parental concern over misinformation about the vaccine and vaccine safety disseminated by a lobby group set up in 2015 [12]. This culminated in a highly publicized anti-vaccine documentary on television, which led to the establishment of a national office in Ireland for responding to and addressing the concerns of groups that opposed vaccination. Moreover, previously printed materials, online messages, etc. about HPV vaccine were revised and additional materials were created to promote the vaccine, which resulted in restoring people’s confidence in HPV vaccination. Thus, it is necessary to improve the knowledge regarding the HPV vaccine for cervical cancer and vaccination health literacy among the younger generation.

The Japanese Ministry of Health, Labour, and Welfare (MHLW, Tokyo, Japan) reinstated the official active recommendation for HPV vaccination in November 2021. Despite this, we believe that unless health literacy about cervical cancer and the HPV vaccine among young women who are eligible for vaccination improves, it is likely that vaccination uptake will remain sluggish [13,14,15]. Moreover, approximately 9 years of declining HPV vaccination rates and public confidence during this period may pose difficulty in increasing the HPV vaccination rate among young women in Japan. Therefore, the causes behind people’s hesitancy toward the HPV vaccine need to be discussed openly and in depth. Moreover, regaining people’s trust in medicine by addressing their concerns and determining effective methods for providing the correct information regarding HPV vaccination is essential. Therefore, we aimed to conduct a randomized study using different methods of providing medical information (print-based and social networking service (SNS)-based educational interventions) to improve health literacy regarding cervical cancer and HPV vaccination among female students in Japan. For the purpose of this study, an SNS-based innovative informatics platform for unvaccinated young women to facilitate accurate HPV vaccination and vaccination completion rates was developed. To the best of our knowledge, this study was the first randomized trial to investigate whether an SNS-based intervention increases knowledge of vaccination and HPV outcomes among young women in Japan, following the discontinuation of HPV vaccine recommendations by the government. In this paper, we present the results of a preliminary study and discuss recent topics related to HPV vaccination in Japan.

## 2. Materials and Methods

### 2.1. Study Settings and Participants

A total of 15,450 female students from 17 departments of institutes of a private university group, including Teikyo University (10,000 students), Teikyo Heisei University (5000 students), and Teikyo Institute of Advanced Nursing (450 students) have been enrolled in the ongoing study (Figure 1). At the time of enrolment, the students were assured that their participation in the study was voluntary and would not affect their academic performance. Furthermore, due to the COVID-19 pandemic, fewer students attended the university, and consequently, 27 female students participated in the preliminary investigation reported in this paper.

The inclusion criteria were young female students (1) aged 18–26 years (2) who had received less than three doses of the HPV vaccine or were unvaccinated, and (3) who could access and use SNS-based programs and follow-up questionnaires. Exclusion criteria were students (1) who faced mental and physical challenges using SNS-based programs and (2) who had completed three doses of the vaccine. All students and their parents provided written informed consent. The study was approved by the institutional review board of Teikyo University. A data monitoring committee was not deemed necessary for this feasibility study as we did not anticipate any adverse events; nonetheless, any unintended consequences of the interventions were recorded.

### 2.2. Study Design

A cluster-randomized parallel-group trial with three groups will be conducted: Group 1, no intervention (control); Group 2, educational intervention (print-based education); and Group 3, educational intervention (SNS-based education). The presence or absence of an educational effect will be determined by comparing Groups 1 and 2 with Group 3 for discussion purposes (Figure 2).

### 2.3. Allocation Method

A faculty or department and the allocation unit are considered as clusters. Allocators prepare an allocation table using a stratified block randomization method with two stratification factors: medical and non-medical institution (Teikyo University (Tokyo, Japan) + Teikyo Institute of Advanced Nursing Studies (Tokyo, Japan) and Teikyo Heisei University (Tokyo, Japan)). This is an open-label study, and both the allocators and participants know the allocation results.

### 2.4. Intervention

The medical information and educational tools on cervical cancer and HPV vaccination developed by the principal investigator, which are distributed to female students every six months, are randomly assigned to three arms: Arm 1, no intervention (control); Arm 2, distributed by mail; and Arm 3, distributed through SNS (LINE, Facebook, and Twitter) on websites (Figure 2). All three arms are followed up for 15 months, during which, each intervention (Arms 2 and 3) is conducted three times at 6 month intervals.

### 2.5. Collection Method

A questionnaire is sent to three randomly allocated groups at the time of enrolment (baseline). The research secretariat delivers the allocation results to the enrollees when sending the baseline questionnaire, either by post in a booklet form or via an SNS. The survey questionnaires are emailed to all the groups, and along with the final questionnaire at 15 months, they receive a reminder to complete the questionnaires. The questionnaires inquire about the participants’ faculty affiliation, age, diet, HPV vaccination history, and whether they had a family health care provider, smoked, exercised voluntarily, ever visited an obstetrician, received routine vaccinations, as well as their knowledge regarding cervical cancer, using the health literacy scale. At the time of questionnaire delivery, students in the no-intervention control group are informed that they will not receive any medical information or educational tools on cervical cancer and HPV vaccination and will be sent the same questionnaire, omitting basic information, thrice during enrolment.

### 2.6. Outcomes

#### 2.6.1. Primary Outcome

The primary outcome is (1) increasing self-reported HPV vaccine uptake rates via SNS-based educational contents: from no intention to at least taking one dose or all three doses and (2) increasing knowledge of susceptibility to cervical cancer, its severity, and benefit from HPV vaccination as measured by the reliable and valid health belief model [16]. 

#### 2.6.2. Secondary Outcome

We investigate vaccination barriers on self-reported HPV vaccine uptake and behavioral intention or attitude changes toward HPV vaccine uptake in support of educational contents between unvaccinated and vaccinated participants who had previously received a single or double dose of the HPV vaccine.

### 2.7. Statistical Analysis

#### Primary Outcome

Participants answer five questions on the communicative and critical health literacy (CCHL) scale ranging from 1 (very easy) to 5 (very difficult) regarding the HPV vaccine and cervical cancer screening. The mean within-individual health literacy score is the mean of the five questions on CCHL; a score of 4 or more is considered “highly health literate.” The primary endpoint is the proportion of participants with high health literacy scores immediately after the third delivery of educational contents (12 months in the study). The null hypothesis is that the proportion of participants with high health literacy in Groups 1 and 2 is equal, while the alternative hypothesis is that the proportion of participants with high health literacy in Group 2 is higher than that in Group 1.

### 2.8. The Target Number of Participants

The proportion of high health literacy under the competing hypotheses was P1 (intervention group) = 0.60 and P2 (control group) = 0.30, implying that the meaningful difference in the proportion of high health literacy between the two groups was 0.30, the within-class correlation was 0.100, the mean cluster size was 200, the two-sided significance level was 0.05, and the power was 0.80. The required number of clusters for each group was calculated as 4.4. If there are five clusters in each group, the total number of required participants is assumed to be 5 × 3 × 200 = 3000, and assuming a dropout rate of about 15%, 3500 participants are required.

### 2.9. Main Analysis

We use a logistic regression mixed-effects model, with the dichotomous variable high or low health literacy as the outcome variable, to calculate the odds ratio of the effect of group on high health literacy with group (control and intervention), institution, medical and non-medical as population effects, and cluster as a variable effect. If the odds ratio is significantly higher than 1, we can conclude that the intervention affected the participants’ health literacy. Adjustment factors such as age and prior knowledge can be included in the statistical model upon the discretion of the statistical analyst and principal investigator. A cluster-randomized study with three arms of groups will be delivered in the following clusters (17 schools): law, economics, literature, foreign language, education, medicine, pharmaceutical sciences (2 schools), medical informatics and technology, nursing (3 schools), science and engineering, modern life, health care, health and medical science, and medical sports.

## 3. Results

### 3.1. Participant Characteristics

We conducted a preliminary investigation between 2019 and 2020 in Japan. Due to the coronavirus disease (COVID-19) pandemic, only a few students (27 applicants) participated in the study, which was conducted in our laboratory at Teikyo University, after seeing the participation invite posted in the teaching department. Information pertaining to students’ faculty affiliation, family background, smoking status, attitudes toward health, and routine vaccination status was collected. Furthermore, the extent of their knowledge regarding cervical cancer and the HPV vaccine was noted. Although one of the inclusion criteria for study participants was being unvaccinated for HPV, six HPV-vaccinated students were included in this preliminary study (Table 1).

### 3.2. Health Literacy Scale

The participants were divided into three randomly assigned groups and sent a questionnaire at enrolment (baseline). As this was a preliminary investigation, the number of participants was small to allow comparison between the three groups; therefore, we summarized the results for all the participants. Two rounds of the survey were conducted, and the preliminary results are shown in Table 2. The participants had no difficulty understanding the questions and answered all of them. Responses to Item 11 in the questionnaire indicated a significant improvement in the participants’ knowledge, that is, malignant findings in the cervix and excision by methods such as conization increase the risk of imminent miscarriage and premature birth (Table 2). Although it is widely believed that early detection of cervical lesions is needed to treat them, surprisingly, the participants were unaware that surgery involving the uterus is associated with perinatal risks.

### 3.3. CCHL Scale

The results of the CCHL scale are presented in Table 3. As this was a preliminary investigation, we ensured that the participants understood all the questions and were able to answer them with ease. Although the number of participants was small for conducting a robust analysis, most students provided a rating of 2 (somewhat easy) in the first and second survey rounds for all the five items on the scale. However, many students responded “very difficult” to Item 3—being able to understand and communicate information related to the HPV vaccine and cervical cancer screening to others. At this stage, the first-round results are not adequate to draw a firm conclusion. Therefore, we plan to conduct a second round of interventions to assess whether the scores would improve further, and if so, whether faculty affiliation would be an influencing factor.

## 4. Discussion

Currently, national HPV vaccination programs are publicly funded in more than 120 countries worldwide [17]. In Japan, public subsidies for HPV vaccination began in 2010, and it became a routine vaccination based on the Immunization Law in April 2013 [18]. However, in June 2013, the Japanese government stopped recommending temporary prophylactic vaccination after symptoms, including chronic pain and motor dysfunction, after HPV vaccination were reported in young women and brought to light by the Japanese media [19,20]. The reason for this might have been that some HPV vaccine recipients suffered from symptoms that could not be ruled out as being related to the vaccination; moreover, there were reports suggesting a causal relationship between the HPV vaccine and severe symptoms such as postural orthostatic tachycardia syndrome (POTS) and chronic regional pain syndrome (CRPS) [21]. However, subsequent investigations have failed to provide any scientific or epidemiological evidence of a causal relationship between the various symptoms reported after vaccination, such as pain and motor dysfunction, and the vaccination [22]. Consequently, the MHLW’s Adverse Effects Review Committee confirmed that the various symptoms reported after vaccination were functional physical symptoms [23,24]. A nationwide epidemiological survey conducted by the Sobue Group of the MHLW reported that a few symptoms similar to those reported as post-vaccination symptoms existed among people without a history of HPV vaccination [25]. In a questionnaire survey of women born between 1994 and 2000 in Nagoya, there was no significant difference in the age-adjusted incidence of 24 symptoms between vaccinated and unvaccinated women and no evidence of a causal relationship between the symptoms and HPV vaccination [26]. The Japanese government’s decision to stop actively recommending HPV vaccination lasted for 8 years, from June 2013 to November 2021. During this period, WHO criticized Japan for exposing young women to the risk of inherently preventable HPV-related cancers [27]. Meanwhile, Australia published a simulation showing that it would become the first country in the world to eliminate new cervical cancer cases by 2028 [28]. In Sweden, HPV vaccination in women aged 10–30 years is expected to significantly reduce the risk of invasive cervical cancer at the national level [29]. Furthermore, in Denmark, which has experienced a temporary drop in HPV vaccination coverage due to the media [30], a study found no causal adverse relationship between the quadrivalent HPV vaccine and possible symptoms [31].

Surveillance after HPV vaccination must continue, but we should also consider the rigorous treatment that patients need to undergo on contracting cervical cancer. The efficacy of HPV vaccine outweighs the risk of adverse symptoms. Therefore, the HPV vaccination rate is on the rise as Japanese MHLW resumed actively recommending the HPV vaccine program in November 2021. In addition, due to the COVID-19 pandemic, people’s physical and emotional barriers to vaccination to prevent transmission of the virus gradually eased [32]. Thus, as the COVID-19 vaccination coverage in Japan is over 70%, more young women are keen to resume getting vaccinated for HPV.

We conducted a preliminary investigation that compared the knowledge of cervical cancer and the HPV vaccine among female students before and after the distribution of print-based and SNS-based educational contents. Due to the small number of students included in the analysis, it was impossible to draw any firm conclusions; however, the results showed a general tendency for improvement in knowledge. Although non-medical students may have found it difficult to understand the educational contents, their participation in the study could have led them to seek more knowledge through the Internet. In the future, we would like to analyze communication failures regarding information on HPV from medical, social-scientific, and behavioral perspectives. Furthermore, participants’ health literacy improved as indicated by the results of the second survey. However, as the data were not adequate, we were unable to compare the three groups and comment on the impact of the educational interventions on knowledge; nonetheless, the motivation to participate in the study alone could likely indicate a behavioral change in the participants.

In the future, we hope to reach our target of 3500 participants. The study aimed to include female students aged 18–26, who missed their routine vaccination periods and were not yet eligible for vaccination after the Japanese MHLW resumed active vaccination. Currently, there is an ongoing debate in Japan about subsidizing catch-up vaccinations, and there has been no decision on introducing the more promising 9-valent vaccine as a routine vaccine. Students who wish to be vaccinated prior to the catch-up vaccination scheme must pay for them; moreover, many of them have been self-inoculated. Therefore, it is crucial that we, especially medical professionals, inform them about routine HPV vaccination and the risk of cervical cancer, which often affects women in their 20s and 30s, based on accurate scientific knowledge. Some studies have analyzed knowledge and awareness about cervical cancer and HPV vaccines in other countries [33,34,35,36,37,38,39,40], but few such studies exist in Japan. This study targeting Japanese female university students who had not received the HPV vaccine aimed to work with the students through intervention and research to determine their views on health care. Moreover, the results of this study could help improve knowledge and awareness of cervical cancer among young women.

## 5. Conclusions

There is a need for continued grassroots efforts to improve knowledge of cervical cancer and HPV vaccination among female students in Japan. Medical professionals, including obstetricians and gynecologists, need to take responsibility and provide accurate scientific knowledge regarding cervical cancer to female students and their parents. Furthermore, efforts need to be made to eradicate cervical cancer. Finally, we continue to recruit participants for this study and hope to publish comprehensive results after completing the study.

## Figures and Tables

**Figure 1 ijerph-19-05191-f001:**
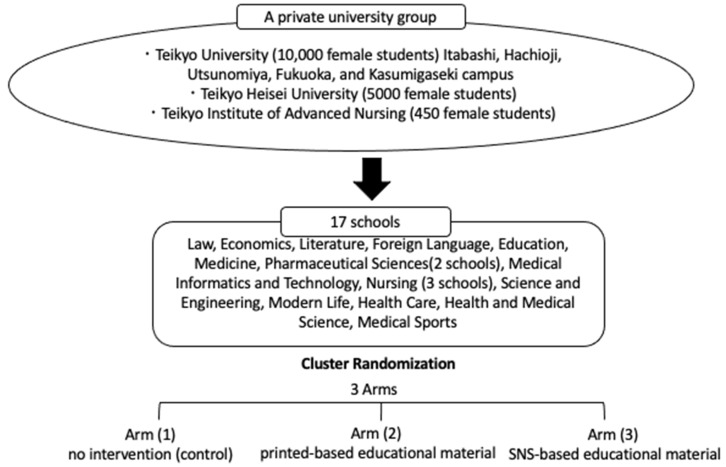
Study flowchart.

**Figure 2 ijerph-19-05191-f002:**
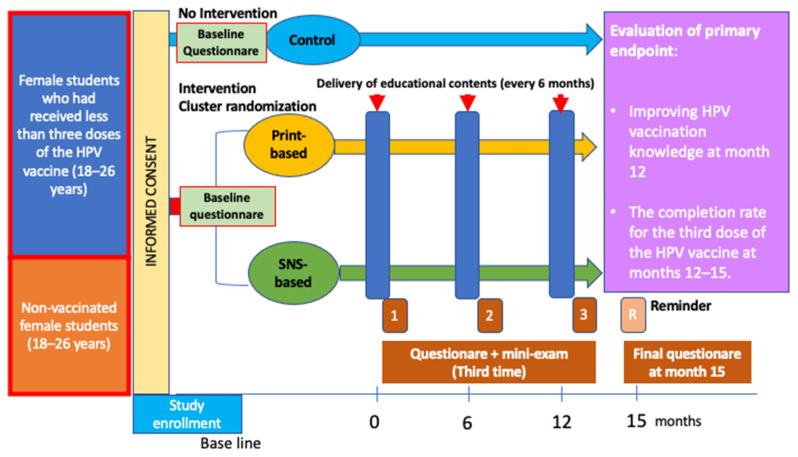
Study procedure.

**Table 1 ijerph-19-05191-t001:** Participant characteristics.

Variable	N	%
Affiliation		
Pharmaceutics	5	18.5
Medicine	6	22.2
Nursing	9	33.3
Literature	3	11.1
Science and technology	1	3.7
Medical technology	2	7.4
Family health care provider	8	29.6
Family background	5	18.5
Smoking status	0	0
Balanced diet	19	70.4
Exercise voluntarily		
Once a week	4	14.8
Twice or three times per week	4	14.8
Every day	3	11.1
Obstetrics and gynecology consultation history	14	51.9
Routine vaccination status	23	85.2
HPV vaccination history		
None	17	63.0
Once	4	14.8
Twice	0	0
Thrice	6	22.2

**Table 2 ijerph-19-05191-t002:** Results of the health literacy scale (first and second survey rounds).

Item	N = 27			
	* N^1st^	%	* N^2nd^	%
(1) Cervical cancer is an infectious disease caused by the human papillomavirus (HPV).	20	74.1	27	100
(2) There are more than 100 types of HPV, of which 14 cause cervical cancer (high-risk HPV).	7	25.9	9	33.3
(3) Persistent infection with high-risk HPV causes cervical cancer.	8	29.6	11	40.6
(4) HPV is ubiquitous and a common virus that can be transmitted to the uterus through a single sexual activity.	21	77.8	25	92.6
(5) Over 80% women with a history of sexually transmitted infection will acquire HPV with age.	5	18.5	8	29.6
(6) HPV can be transmitted to women and men.	15	55.6	18	66.7
(7) Cervical cancer is the most common cancer among women in their 20s and 30s.	15	55.6	20	74.1
(8) Approximately 3000 patients pass away from cervical cancer every year in Japan.	10	37.0	12	44.4
(9) There is possible delayed detection of cervical cancer even after having annual check-ups.	14	51.9	17	63.0
(10) Even if cervical cancer is detected in an early stage, removing the uterus is necessary.	11	40.7	18	66.7
(11) Even if an abnormality is detected prior to having cervical cancer, part of the uterus, which may cause premature delivery, needs to be removed.	6	22.2	14	51.9
(12) HPV vaccines that can prevent cervical cancer exist.	23	85.2	27	100
(13) There is significant evidence that HPV vaccines can prevent cervical cancer.	24	88.9	27	100
(14) The HPV vaccine is more effective in preventing cervical cancer when it is administered prior to acquiring HPV (before sexual intercourse).	20	74.1	20	74.1
(15) Even after getting vaccinated, regular checkups for early detection of cervical cancer are required as there is possibility of getting infected.	17	63.0	21	77.8
(16) Receiving “catch-up vaccination” after the generation of HPV is vaccinated is recommended.	7	25.9	12	44.4
(17) The adverse reactions to the HPV vaccine drew media attention 5 years ago in Japan.	22	81.5	24	88.9
(18) No evidence or cause of adverse reactions to HPV vaccines has been found in Japan.	10	37.0	13	48.1
(19) There is scientific evidence about efficiency and safety of the HPV vaccine.	13	48.1	17	63.0
(20) The Japan Society of Obstetrics and Gynecology strongly urges reinstating the active recommendation of HPV vaccines.	14	51.9	19	70.1

* The number of participants in each survey round.

**Table 3 ijerph-19-05191-t003:** Results of the communicative and critical health literacy scale (first and second survey rounds).

Item		Degree of Difficulty			N (%)	(N = 27)
	Very Easy	Slightly Easy	Intermediate	Slightly Difficult	Very Difficult	Not Applicable
(1) You can gather information related to the HPV vaccine and cervical cancer screening from various sources, such as newspapers, books, television, and the Internet.	1st					
	12 (44.4)	10 (58.8)	3 (11.1)	2 (7.4)	0 (0)	0 (0)
	2nd					
	8 (29.6)	12 (44.4)	4 (14.8)	3 (11.1)	0 (0)	0 (0)
(2) You can select the relevant information from a large amount of information related to the HPV vaccine and cervical cancer screening.	1st					
	2 (7.4)	10 (58.8)	7 (25.9)	7 (25.9)	1 (3.7)	0 (0)
	2nd					
	5 (18.5)	10 (58.8)	4 (14.8)	7 (25.9)	1 (3.7)	0 (0)
(3) You can understand and communicate to others information related to the HPV vaccine and cervical cancer screening.	1st					
	0 (0)	6 (22.2)	8 (29.6)	8 (29.6)	4 (14.8)	0 (0)
	2nd					
	1 (3.7)	11 (40.7)	8 (29.6)	3 (11.1)	3 (11.1)	0 (0)
(4) You can determine the reliability of the information related to the HPV vaccine and cervical cancer screening	1st					
	0 (0)	4 (14.8)	9 (33.3)	12 (44.4)	2 (7.4)	0 (0)
	2nd					
(5) You can develop plans and actions to improve your health based on the information related to the HPV vaccine and cervical cancer screening.	0 (0)	10 (58.8)	7 (25.9)	6 (22.2)	4 (14.8)	0 (0)
	1st					
	1 (3.7)	6 (22.2)	11 (40.7)	9 (33.3)	0 (0)	0 (0)
	2nd					
	3 (11.1)	12 (44.4)	7 (25.9)	3 (11.1)	2 (7.4)	0 (0)

## Data Availability

The data presented in this study are available on request from the corresponding author.

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
