# Peer review of "Effect of Different Educational Interventions on Knowledge of HPV Vaccination and Cervical Cancer among Young Women: A Preliminary Report"

_ijerph, 2022, doi:10.3390/ijerph19095191_

Round 1

Reviewer 1 Report

The topic of this manuscript falls within the scope of International Journal of Environmental Research and Public Health.

The aim of the study was to conduct a randomized study using different methods of providing educational contents to improve health literacy regarding cervical cancer and HPV vaccination among female students in Japan. A total of 15,450 female students have been enrolled in the study. The Authors showed that participants’ knowledge and health literacy improved post intervention. Therefore, they concluded that medical professionals must provide accurate scientific knowledge regarding routine HPV vaccination and the risk of cervical cancer to young women to improve their health literacy and subsequently increase the HPV vaccination rates.

the strength of this paper: very interesting topic, introduction-relevant and concise; material and methods-the right choice of methodology methods, which was presented in comprehensible way; the obtained results are presented in the form of figures and tables, which are clear and easy to understand; the discussion- supports the results properly and refers to the current literature in appropriate manner; the conclusions- based on the obtained results.

The topic of this manuscript falls within the scope of International Journal of Environmental Research and Public Health. In my opinion the topic of presented paper is original and very important taking into account ani increasing morbidity of cervical cancer in females.

The main question addressed by the research is whether an social networking service (SNS)-based intervention increases knowledge of vaccination and HPV outcomes among young women in Japan following the discontinuation of HPV vaccine recommendations by the                         government. This study is the first randomized trial, which investigate it.

The aim of the study was to conduct a randomized study using different methods of providing educational contents to improve health literacy regarding cervical cancer and HPV vaccination among female students in Japan. A total of 15,450 female students have been enrolled in the study. The Authors showed that participants’ knowledge and health literacy improved post intervention. Therefore, they concluded that medical professionals must provide accurate scientific knowledge regarding routine HPV vaccination and the risk of cervical cancer to young women to improve their health literacy and subsequently increase the HPV vaccination rates.

the strength of this paper: very interesting topic, introduction-relevant and concise; material and methods-the right choice of methodology methods, which was presented in comprehensible way; the obtained results are presented in the form of figures and tables, which are clear and easy to understand; the discussion- supports the results properly and refers to the current literature in appropriate manner; the conclusions- based on the obtained results, they are consistent with evidence and arguments. They address the main question posed.  The Authors used appropriate references.

Author Response

The topic of this manuscript falls within the scope of International Journal of Environmental Research and Public Health.

The aim of the study was to conduct a randomized study using different methods of providing educational contents to improve health literacy regarding cervical cancer and HPV vaccination among female students in Japan. A total of 15,450 female students have been enrolled in the study. The Authors showed that participants’ knowledge and health literacy improved post intervention. Therefore, they concluded that medical professionals must provide accurate scientific knowledge regarding routine HPV vaccination and the risk of cervical cancer to young women to improve their health literacy and subsequently increase the HPV vaccination rates.

the strength of this paper: very interesting topic, introduction-relevant and concise; material and methods-the right choice of methodology methods, which was presented in comprehensible way; the obtained results are presented in the form of figures and tables, which are clear and easy to understand; the discussion- supports the results properly and refers to the current literature in appropriate manner; the conclusions- based on the obtained results.

The topic of this manuscript falls within the scope of International Journal of Environmental Research and Public Health. In my opinion the topic of presented paper is original and very important taking into account ani increasing morbidity of cervical cancer in females.

The main question addressed by the research is whether an social networking service (SNS)-based intervention increases knowledge of vaccination and HPV outcomes among young women in Japan following the discontinuation of HPV vaccine recommendations by the government. This study is the first randomized trial, which investigate it.

The aim of the study was to conduct a randomized study using different methods of providing educational contents to improve health literacy regarding cervical cancer and HPV vaccination among female students in Japan. A total of 15,450 female students have been enrolled in the study. The Authors showed that participants’ knowledge and health literacy improved post intervention. Therefore, they concluded that medical professionals must provide accurate scientific knowledge regarding routine HPV vaccination and the risk of cervical cancer to young women to improve their health literacy and subsequently increase the HPV vaccination rates.

the strength of this paper: very interesting topic, introduction-relevant and concise; material and methods-the right choice of methodology methods, which was presented in comprehensible way; the obtained results are presented in the form of figures and tables, which are clear and easy to understand; the discussion- supports the results properly and refers to the current literature in appropriate manner; the conclusions- based on the obtained results, they are consistent with evidence and arguments. They address the main question posed.  The Authors used appropriate references.

⇒ Thank you for your positive comments.

Reviewer 2 Report

The manuscript entitled "Effect of different educational interventions on knowledge of HPV vaccination and cervical cancer among young female adults: A preliminary report, provides data on a feasibility study to assess two different formats of HPV vaccination education directed at 16 to 18 year old University students. Due to various issues with study participant recruitment, including the COVID pandemic  - only 27 female students were recruited, therefore no firm conclusions can be drawn from the study results.  The authors need to decide what the value of reporting their study is, despite the limited inconclusive data.  The methodology is reported very well and the details of their questionnaires and educational material is useful to know.

My main issue with the study is that the authors minimize concerns about vaccine adverse effects and assumed vaccine hesitancy is due to "misinformation".  They do not state exactly what the "misinformation" is or tackle each component of the alleged "misinformation" specifically.  They assume that if people know the risks of developing cervical cancer and the effectiveness of the vaccines at preventing cervical cancer, vaccine hesitance will be reduced and uptake will increase.  I believe this is a rather naive view of the issue.  The authors do not properly investigate the reported side-effects or the new reports and/or case-studies in Japan that specifically led to a drastic decline in vaccine uptake.  The authors also assume that, because COVID vaccine uptake has been high in Japan, people are now more likely to take up the HPV vaccination, but we have yet to see what increasing reports of COVID vaccine side-effects and the media debates surrounding them will do in the future.

Here are specific critiques to specific segments of the manuscript: - 

Line 18:  The authors state that one of the aims of the manuscript, given the preliminary nature of the findings is to "discuss current topics related to HPV vaccination in Japan" - I think, they can greatly improve the manuscript if they do more justice to this discussion.  What "misinformation" is out there and what evidence is there to counter this "misinformation".  There is no "smoke without a fire" and the medical field / vaccine scientists would do better if they could be more honest and transparent about vaccine adverse effects.  There are various critical issues with how vaccine adverse effects are reported and verified. 

The arguments for the risk of death and severe illness if/when cervical cancer is contracted are clearly very strong but vaccine informed consent should be based on a risk/benefit ratio.  This is a critical ethical issue. We need to very systematically and clearly present the arguments for the benefits of the vaccine versus the risk of adverse effects.  Very few scientific references were presented and discussed specifically on the topic of HPV vaccine-associated adverse effects.

Manuscript Line or segment specific critiques: - 

  • Line 20:  define SNS fully within the abstract.  It is the first-mention of the acronym and it is not fully define i.e., Social Networking Services.
  • Line 67:  "misinformation about the vaccine disseminated by a lobby group set up in 2015 in Ireland" is cited as the reason vaccine hesitancy declined drastically in Japan.  One reference is provided for this - a  short correspondence in the lancet that itself only provides two references and dismisses the mothers concerns.  It does not delve into the case studies or address specific content of the documentary it dubs as "misinformation".  This is not an evidence-based statement and the specific side effects and case studies of concern should be reviewed in a systemic, unbiased fashion if the topic is to be addressed properly.
  • Lines 75 and 76: Again as above, the general statement is made that "..approximately 9 years of declining HPV vaccination rates and public confidence during this period may pose difficulty in increasing the HPV vaccination rate among young female adults in Japan".  As indicated above, while this manuscript is attempting to show that their specific approach may work as a tool to increase "medical literacy", I feel the authors are missing the main issue that, it is not the vaccine efficacy worrying the general population but the adverse effects.  The case-studies and word-of-mouth stories that have scared the public away from the vaccine need to be addressed in an open and honest, rather than dismissive way, if the activities that are planned as a follow-up to this study are to have a long lasting effect.  The issue isn't so much educating the public and combating vaccine hesitancy and misinformation in my view but in winning back trust in the medical community and addressing public concerns effectively.
  • Line 230 to 235: states that "The government stopped recommending the vaccine in June 2013 due to various adverse effects that were brought to light by the media."  This indicates that there was a real and substantial issue.  The discussion should delve into this more.  What exactly were the side-effects reported.  What are the risks of these side effects occurring in the general population and how sound are the studies that estimate these risks.  How reliable is the data disproving these effects.
    • Generally speaking, authors provide only 1 or 2 references to support their claims and do not do a thorough enough investigation into the background issues underlying their study so they do not provide confidence that their intervention will adequately or successfully tackle the issue.
  • Line 241:  E.g. "A survey of adverse effects was conducted in 1994 to 2000 and in Nagoya and there was no significant differences in age-adjusted incidence of 24 symptoms between the vaccinated and unvaccinated."  What is the strength of this study - it is a survey in one city, completed 13 years before the government stopped recommending vaccination, so there must have been other, more substantial information that came to light, that caused the government to stop recommending the vaccination.  What was the vaccination rate in 1994 to 2000 compare to 2013?  What further evidence came to light after 2000?
  • Line 251 and 252: "a (singular) study found no causal adverse relationship between the quadrivalent HPV vaccine and possible symptoms.  This is a single population-based retrospective cohort study conducted in Denmark in 2019. 
    • The authors do not do a comprehensive enough literature review on the topic.  I found a series of articles via this search https://pubmed.ncbi.nlm.nih.gov/28730271/... 
    • This specific example that came up in the search for example - 10.1007/s12026-018-9036-1 - by Blitshteyn at al. in 2018 proposes that "vaccine-triggered immune-mediated automic dysfunction could lead to the development of de novo post-HPV vaccination syndrome possibly in genetically susceptible individuals" and calls for "well-designed case-control studies to determine the prevalence and possible causation between these symptom clusters and HPV vaccines."
  • Line 253:  The authors state that "The efficacy of HPV vaccine outweighs the risk of adverse symptoms".  They do not provide any references or any strong systematic evidence to support this statement.  Most post-surveillance data on vaccine side-effects relies on passive reporting systems that have been shown to be flawed and to under-report the events (see https://pubmed.ncbi.nlm.nih.gov/17712091/).  Even the WHO admitted in conference meetings before the COVID pandemic that the issue of poor post-vaccination surveillance needs to be addressed.  So there is a wider issue that the manuscript is ignoring.
  • Line 261 to 262: points to the weakness of the results but indicates that the methodology could work.  I think social media can be used effectively to educate but the authors need to tackle exactly what education is needed.  Sound evidence-based information is required and the authors have not tackled this well enough.
  • Lines 263 - 264:  "non-medical students found it difficult to understand the edducational contents".  I think this is another important issue that the study team needs to address if it is to be successful in any interventions that proceed from this.  Rather than tackling the issue from solely a medical perspective, a social-science or behavioral analysis of the communication failures could be considered.  Effective science-communication to a lay-audience in the face of strong anti-vaccine lobbies, is something that government health ministries need to address. I don't think the authors are addressing the issue deeply enough.  Other countries have used social media effectively to combat vaccine hesitancy but we need to be careful that we are not countering what we dismiss as "misinformation" with more "misinformation".
  • Line 275 - 276:  There are studies to indicate that the nonivalent vaccine is associated with more adverse effects.  According to this paper - https://link.springer.com/article/10.1007/s10067-017-3768-5 -  by Martinez-Lavin & Amezcua-Guerra " Two of the largest randomized HPV vaccine trials unveiled more severe adverse events in the tested HPV vaccine arm of the study. Nine-valent HPV vaccine has a worrisome number needed to vaccinate/number needed to harm quotient. Pre-clinical trials and post-marketing case series describe similar post-HPV immunization symptoms."  This and other similar studies should be cited and analyzed in a unbiased systematic fashion and the Japanese and other governments make critical decisions about this evidence.  They should provide clear, accurate informed consent to their population.  The information should be understandable to a lay audience.  This is a critical ethical issue.

    Author Response

    Dear reviewer,

    Thank you very much for your valuable comments.

    We have replied the comments as shown below.

    Sincerely,

    Kazunori Nagasaka

    The manuscript entitled "Effect of different educational interventions on knowledge of HPV vaccination and cervical cancer among young female adults: A preliminary report, provides data on a feasibility study to assess two different formats of HPV vaccination education directed at 16 to 18 year old University students. Due to various issues with study participant recruitment, including the COVID pandemic  - only 27 female students were recruited, therefore no firm conclusions can be drawn from the study results.  The authors need to decide what the value of reporting their study is, despite the limited inconclusive data.  The methodology is reported very well and the details of their questionnaires and educational material is useful to know.

    My main issue with the study is that the authors minimize concerns about vaccine adverse effects and assumed vaccine hesitancy is due to "misinformation".  They do not state exactly what the "misinformation" is or tackle each component of the alleged "misinformation" specifically.  They assume that if people know the risks of developing cervical cancer and the effectiveness of the vaccines at preventing cervical cancer, vaccine hesitance will be reduced and uptake will increase.  I believe this is a rather naive view of the issue.  The authors do not properly investigate the reported side-effects or the new reports and/or case-studies in Japan that specifically led to a drastic decline in vaccine uptake.  The authors also assume that, because COVID vaccine uptake has been high in Japan, people are now more likely to take up the HPV vaccination, but we have yet to see what increasing reports of COVID vaccine side-effects and the media debates surrounding them will do in the future.

    Here are specific critiques to specific segments of the manuscript: - 

    Line 18:  The authors state that one of the aims of the manuscript, given the preliminary nature of the findings is to "discuss current topics related to HPV vaccination in Japan" - I think, they can greatly improve the manuscript if they do more justice to this discussion.  What "misinformation" is out there and what evidence is there to counter this "misinformation".  There is no "smoke without a fire" and the medical field / vaccine scientists would do better if they could be more honest and transparent about vaccine adverse effects.  There are various critical issues with how vaccine adverse effects are reported and verified. 

    The arguments for the risk of death and severe illness if/when cervical cancer is contracted are clearly very strong but vaccine informed consent should be based on a risk/benefit ratio.  This is a critical ethical issue. We need to very systematically and clearly present the arguments for the benefits of the vaccine versus the risk of adverse effects.  Very few scientific references were presented and discussed specifically on the topic of HPV vaccine-associated adverse effects.

    ⇒Thank you for your important comments. This study did not aim to review the risks and benefits of the HPV vaccine, rather it explores how to effectively communicate the risks and benefits to young people through research. Nonetheless, we will fully examine your comments and address them in our future research.

    Manuscript Line or segment specific critiques: - 

    • Line 20:  define SNS fully within the abstract.  It is the first-mention of the acronym and it is not fully define i.e., Social Networking Services.

    ⇒ Thank you for pointing this out. The required correction has been made in the Abstract.

    • Line 67:  "misinformation about the vaccine disseminated by a lobby group set up in 2015 in Ireland" is cited as the reason vaccine hesitancy declined drastically in Japan.  One reference is provided for this - a  short correspondence in the lancet that itself only provides two references and dismisses the mothers concerns.  It does not delve into the case studies or address specific content of the documentary it dubs as "misinformation".  This is not an evidence-based statement and the specific side effects and case studies of concern should be reviewed in a systemic, unbiased fashion if the topic is to be addressed properly.

    ⇒ Thank you for your comments. We have described in detail what happened in Ireland in 2015, highlighting how people’s concerns were addressed (Lines 70–75).

    • Lines 75 and 76: Again as above, the general statement is made that "..approximately 9 years of declining HPV vaccination rates and public confidence during this period may pose difficulty in increasing the HPV vaccination rate among young female adults in Japan".  As indicated above, while this manuscript is attempting to show that their specific approach may work as a tool to increase "medical literacy", I feel the authors are missing the main issue that, it is not the vaccine efficacy worrying the general population but the adverse effects.  The case-studies and word-of-mouth stories that have scared the public away from the vaccine need to be addressed in an open and honest, rather than dismissive way, if the activities that are planned as a follow-up to this study are to have a long lasting effect.  The issue isn't so much educating the public and combating vaccine hesitancy and misinformation in my view but in winning back trust in the medical community and addressing public concerns effectively.

    ⇒ We agree with your assessment. As the goal of this study was not to describe the risks and benefits of the HPV vaccine, but rather why such a study is necessary, we have revised the text to indicate how the correct information should be communicated (Lines 84–88).

    • Line 230 to 235: states that "The government stopped recommending the vaccine in June 2013 due to various adverse effects that were brought to light by the media."  This indicates that there was a real and substantial issue.  The discussion should delve into this more.  What exactly were the side-effects reported.  What are the risks of these side effects occurring in the general population and how sound are the studies that estimate these risks.  How reliable is the data disproving these effects.
      • Generally speaking, authors provide only 1 or 2 references to support their claims and do not do a thorough enough investigation into the background issues underlying their study so they do not provide confidence that their intervention will adequately or successfully tackle the issue.
    • Line 241:  E.g. "A survey of adverse effects was conducted in 1994 to 2000 and in Nagoya and there was no significant differences in age-adjusted incidence of 24 symptoms between the vaccinated and unvaccinated."  What is the strength of this study - it is a survey in one city, completed 13 years before the government stopped recommending vaccination, so there must have been other, more substantial information that came to light, that caused the government to stop recommending the vaccination.  What was the vaccination rate in 1994 to 2000 compare to 2013?  What further evidence came to light after 2000?

       ⇒Thank you for your comments. As written in the text, the survey has been conducted for women born between 1994 and 2000 in Nagoya. No specific evidence related to vaccination has occurred around the period.

    • Line 251 and 252: "a (singular) study found no causal adverse relationship between the quadrivalent HPV vaccine and possible symptoms.  This is a single population-based retrospective cohort study conducted in Denmark in 2019. 
      • The authors do not do a comprehensive enough literature review on the topic.  I found a series of articles via this search https://pubmed.ncbi.nlm.nih.gov/28730271/... 
      • This specific example that came up in the search for example - 10.1007/s12026-018-9036-1 - by Blitshteyn at al. in 2018 proposes that "vaccine-triggered immune-mediated automic dysfunction could lead to the development of de novo post-HPV vaccination syndrome possibly in genetically susceptible individuals" and calls for "well-designed case-control studies to determine the prevalence and possible causation between these symptom clusters and HPV vaccines."
    • Line 253:  The authors state that "The efficacy of HPV vaccine outweighs the risk of adverse symptoms".  They do not provide any references or any strong systematic evidence to support this statement.  Most post-surveillance data on vaccine side-effects relies on passive reporting systems that have been shown to be flawed and to under-report the events (see https://pubmed.ncbi.nlm.nih.gov/17712091/).  Even the WHO admitted in conference meetings before the COVID pandemic that the issue of poor post-vaccination surveillance needs to be addressed.  So there is a wider issue that the manuscript is ignoring.
    • Line 261 to 262: points to the weakness of the results but indicates that the methodology could work.  I think social media can be used effectively to educate but the authors need to tackle exactly what education is needed.  Sound evidence-based information is required and the authors have not tackled this well enough.
    • Lines 263 - 264:  "non-medical students found it difficult to understand the edducational contents".  I think this is another important issue that the study team needs to address if it is to be successful in any interventions that proceed from this.  Rather than tackling the issue from solely a medical perspective, a social-science or behavioral analysis of the communication failures could be considered.  Effective science-communication to a lay-audience in the face of strong anti-vaccine lobbies, is something that government health ministries need to address. I don't think the authors are addressing the issue deeply enough.  Other countries have used social media effectively to combat vaccine hesitancy but we need to be careful that we are not countering what we dismiss as "misinformation" with more "misinformation".
    • Line 275 - 276:  There are studies to indicate that the nonivalent vaccine is associated with more adverse effects.  According to this paper - https://link.springer.com/article/10.1007/s10067-017-3768-5 -  by Martinez-Lavin & Amezcua-Guerra " Two of the largest randomized HPV vaccine trials unveiled more severe adverse events in the tested HPV vaccine arm of the study. Nine-valent HPV vaccine has a worrisome number needed to vaccinate/number needed to harm quotient. Pre-clinical trials and post-marketing case series describe similar post-HPV immunization symptoms."  This and other similar studies should be cited and analyzed in a unbiased systematic fashion and the Japanese and other governments make critical decisions about this evidence.  They should provide clear, accurate informed consent to their population.  The information should be understandable to a lay audience.  This is a critical ethical issue.

    ⇒Thank you for your comments, which we believe are concerning the risks of the HPV vaccine. This study did not examine the risks, rather in this paper, we presented the preliminary results of the study on how to communicate correct information to young people. As for the risks of the HPV vaccine in Japan, we referenced Hineno et al.’s study to support the text.

    Reviewer 3 Report

    The manuscript presents a very important piece of knowledge and is discussing an up-to-date topic. The manuscript title and the abstract are clear. However, the abstract could benefit if the methods will be described in greater details - please include the study design here.

    The introduction part is well-written and provides an understanding of the study rationale.

    Methods - Lines 91-95 - Why out of thousands of enrolled participants, only 27 took part in the study? This is not clear. There is a need to describe the study design before the description of the enrollment. This would make the process more clear.

    The methods' section should include the ethical statement.

    The study results are well presented. A figure could improve the comprehension of the results.

    In the discussion part the study's strength and limitations, as well as the study results implication, must be mentioned.

    Author Response

    Dear Reviewer,

    Thank you very much for your valuable comments. We have replied to the comments as shown below.

    Sincerely,

    Kazunori Nagasaka

    The manuscript presents a very important piece of knowledge and is discussing an up-to-date topic. The manuscript title and the abstract are clear. However, the abstract could benefit if the methods will be described in greater details - please include the study design here.

    The introduction part is well-written and provides an understanding of the study rationale.

    Methods - Lines 91-95 - Why out of thousands of enrolled participants, only 27 took part in the study? This is not clear. There is a need to describe the study design before the description of the enrollment. This would make the process more clear.

    ⇒Thank you for your comment. We have added a note in the methodology section as to why only 27 people participated in this preliminary study (Lines 104–107).

    The methods' section should include the ethical statement.

    ⇒In line 114, we state, “The study was approved by the institutional review board of Teikyo University.”

    The study results are well presented. A figure could improve the comprehension of the results.

    ⇒We have not included a figure because the results are still preliminary; however, we would consider including it after conducting the final analysis.

    In the discussion part the study's strength and limitations, as well as the study results implication, must be mentioned.

    ⇒We have not explored the implications as yet because the results are still preliminary. 

    Round 2

    Reviewer 2 Report

    The paragraphs that they have added to the manuscript to address the concerns I mentioned are less dismissive of valid concerns about vaccine adverse effects and improve the manuscript considerably.